# Relationship between physical activity and ankle osteoarthritis: Implications for metabolic diseases

**Woo Sub Kim[1], Hee-jin Yang[2], Ji Hye Choi[3], Hee Soo Han[2], Dong Yeon Lee[4], Kyoung Min Lee**[2]*

**1** Department of Orthopaedic Surgery, Konkuk University Medical Center, Seoul, Republic of Korea, **2** Department of Orthopaedic Surgery, Seoul National University College of Medicine, Seoul National University Bundang Hospital, Seongnam-si, Gyeonggi, Korea, **3** Department of Orthopaedic Surgery, Korea University Anam Hospital, Seoul, South Korea, **4** Department of Orthopedic Surgery, Seoul National University College of Medicine, Seoul National University Hospital, Seoul, Korea

* oasis100@empal.com

## Abstract

### Background

Regular physical activity is associated with lower cardiometabolic risk, yet musculoskeletal disorders such as ankle osteoarthritis (OA) may limit activity. This study investigated the association between radiographic ankle OA severity and habitual physical activity quantified using MET-min/week.

### Methods

In this retrospective cross-sectional study, consecutive patients evaluated for ankle OA at a tertiary referral foot and ankle clinic (Takakura stage ≥2) between June 2022 and October 2024 were included. Weightbearing ankle radiographs were used to classify OA severity (modified Takakura stage). Physical activity was assessed using the International Physical Activity Questionnaire–Short Form (IPAQ-SF; vigorous, moderate, walking, and total MET-min/week). Patient-reported outcomes were assessed using the Foot and Ankle Outcome Score (FAOS). Correlations between MET-min/week and clinical parameters were evaluated using multivariable partial-correlation analyses controlling for age, sex, BMI, and pain severity (VAS), with adjustment for multiple comparisons.

### Results

A total of 262 patients (99 males, 163 females; mean age 66.8 years) were analyzed. Mean BMI was 26.3 kg/m². Takakura stages were: stage 2 (n = 54), stage 3a (n = 93), stage 3b (n = 52), and stage 4 (n = 63). Moderate activity MET and walking MET differed between males and females (p = 0.033 and p = 0.035, respectively). In the overall cohort, higher age correlated with lower vigorous activity MET (r=−0.155,

**Data availability statement:** If the data are all contained within the manuscript and/or Supporting information files, enter the following: All relevant data are within the manuscript and its Supporting information files.

**Funding:** The author(s) received no specific funding for this work.

**Competing interests:** The authors have declared that no competing interests exist.

p = 0.010) and total MET (r=−0.182, p = 0.003), and higher BMI correlated with lower vigorous MET (r=−0.129, p = 0.038), walking MET (r=−0.141, p = 0.024), and total MET (r=−0.177, p = 0.005). After adjustment for covariates, higher Takakura stage correlated with lower vigorous MET (r=−0.152, p = 0.016) and lower total MET (r=−0.127, p = 0.044). In sex-stratified analyses, Takakura stage correlated with vigorous MET in males (r=−0.226, p = 0.027) and with total MET in females (r=−0.167, p = 0.037). In Takakura stage 4, no patients reported vigorous activity, indicating a floor effect.

## Conclusions

Demographic factors (age, BMI, sex) and radiographic severity of ankle OA were associated with lower self-reported physical activity (MET-min/week). Longitudinal studies incorporating objective activity monitoring and cardiometabolic endpoints are needed to clarify temporal relationships and potential metabolic implications.

## Level of evidence

Level III, retrospective cross-sectional study.

## Introduction

Osteoarthritis is the leading cause of disability in the United States and has been associated with a decrease in health-related quality of life, including mental health, physical function, and general health scores [1]. Approximately 1% of the global population suffers from ankle arthritis, with around 50,000 new cases diagnosed annually [2]. Unlike hip or knee osteoarthritis, is most commonly post-traumatic, developing after prior ankle fractures or ligament injuries [3,4]. Mechanical factors such as joint incongruity, instability, and malalignment contribute to disease onset and progression and may substantially restrict mobility and physical activity [3]. Ankle osteoarthritis is also commonly linked to muscle wasting and a range of neurological and psychological consequences, including altered gait patterns, reduced range of motion, and impaired balance control, all of which contribute to decreased physical activity [5].

Regular physical activity is well established to be associated with a decreased risk of developing metabolic diseases through multiple physiological mechanisms [6]. Engaging in exercise enhances insulin sensitivity, facilitates glucose uptake by skeletal muscle, and improves lipid metabolism [7]. Moreover, physical activity could decrease pro-inflammatory cytokines and increase anti-inflammatory mediators, thereby mitigating chronic low-grade inflammation a key factor in the pathogenesis of metabolic disorders such as type 2 diabetes and cardiovascular disease [8,9].

Measuring physical activity using MET-minutes (Metabolic Equivalent Task-minutes) could provide crucial insights into the functional capacity of individuals with OA, helping clinicians assess their physical activity levels and identify limitations [10].

The International Physical Activity Questionnaire (IPAQ) is a commonly used tool for calculating MET-minutes, quantifying patients' activity levels, providing a metric that is useful for correlating physical activity with the patients' health conditions [11].

The association between physical activity and ankle osteoarthritis as a potential mediator for the development of metabolic diseases has not been extensively studied. The objective of this study was to examine the association between radiographic ankle osteoarthritis severity (Takakura stage) and habitual physical activity quantified using METs (Metabolic Equivalent of Task), and to explore how these relationships vary by demographic and clinical characteristics. We hypothesized that greater ankle osteoarthritis severity would be associated with lower habitual physical activity, and that reduced activity may be relevant to cardiometabolic health.

## Materials and methods

### Ethical consideration and study participants

This study was approved by the Institutional Review Board of our hospital (IRB No. B-2402-880-104), and the requirement for informed consent was waived due to the retrospective nature of the study. The dataset was accessed for research purposes on 19 January 2024. All data were de-identified prior to analysis, and investigators did not have access to any information that could directly identify individual participants.

We conducted a retrospective review of patients who visited our hospital from June 2022 to October 2024. Consecutive patients who visited our foot and ankle clinic (a tertiary referral center) for ankle osteoarthritis were included. These patients underwent weightbearing ankle X-rays and Takakura stage was evaluated. Foot and ankle outcome score (FAOS) and The International Physical Activity Questionnaire Short Form (IPAQ-SF) were collected for all patients with ankle osteoarthritis. Demographic data, including age, sex, and BMI, were also recorded. Exclusion criteria were as follows: 1) patients with previous ankle surgeries, 2) those with neuromuscular diseases, 3) those with ankle osteoarthritis with valgus deformity, 4) those with inflammatory arthritis such as rheumatoid arthritis, 5) those with avascular necrosis around the ankle joint, 6) those who could not ambulate independently, and 7) those with defective data. In addition, Takakura stage 1 cases (n = 8) were excluded due to insufficient sample size and the early-stage nature of the disease, which may not reflect meaningful variability in physical activity levels.

### Takakura staging on the plain radiography

Weightbearing anteroposterior (AP) and lateral views of the ankle were obtained using a UT 2000 X-ray machine (Philips Research, Eindhoven, the Netherlands) at a source-to-image distance of 200 cm and set to 50 kVp and 5 mAs. Radiographic analysis was conducted using a Picture Archiving and Communication System (PACS) software package (Infinitt, Seoul, South Korea). The modified Takakura classification was adopted to evaluat the degree of ankle osteoarthritis, which is a widely used system to grade the severity of medial ankle osteoarthritis and guide treatment decisions: In Stage 1, there is no narrowing of the joint space, but early signs like sclerosis and osteophyte formation are present. Stage 2 shows narrowing of the medial joint space, indicating progression of cartilage loss. In Stage 3, the medial joint space is completely obliterated, leading to direct bone contact; Stage 3a involves bone contact limited to the medial malleolus, while Stage 3b extends to the talar dome. Finally, Stage 4 is characterized by the complete obliteration of the entire joint space with full bone contact, representing advanced osteoarthritis [12]. Radiographic evaluations were performed by a fellowship-trained orthopedic surgeon with eight years of clinical experience in orthopedic surgery.

### Foot and Ankle Outcome Score (FAOS)

The Foot and Ankle Outcome Score (FAOS) questionnaire was used to assess patients' subjective experience of their foot and ankle problems. The FAOS is a region-specific outcome instrument comprised of 42 questions organized within

five subscales: pain, other symptoms, activities of daily living (ADL), sports and recreational activities (Sport/Rec), and foot- and ankle-related quality of life (QoL). Each patient responded to each question on a scale from 0 to 4, and a score for each subscale was calculated according to a unique formula designed for this instrument. The subscale scores range from 0 (extreme symptoms) to 100 (no symptoms). The Korean version of the FAOS has also been cross-culturally adapted and validated to ensure its reliability and applicability in Korean clinical settings, making it a valuable tool for both clinical practice and research in different populations [13]. Each patient was asked to consider symptoms during the past week when completing the questionnaire.

### International Physical Activity Questionnaire (IPAQ)

The International Physical Activity Questionnaire Short Form (IPAQ-SF) was developed as a standardized tool to measure physical activity across different populations. It includes nine questions that ask respondents to recall their physical activity over the past seven days, focusing on four levels of intensity: vigorous activity, moderate activity, walking, and sitting. Vigorous activities include exercises like running, aerobics, or fast cycling, and moderate activities include exercises like leisure cycling, gardening, or brisk walking. Respondents are asked to report how many days per week and how many minutes per day they engaged in these activities. The IPAQ-SF calculates METs for each level of activity intensity using the following formulas. The IPAQ-SF calculates MET-minutes per week for each activity type (vigorous, moderate, walking) by multiplying the respective MET value (8.0 for vigorous activity, 4.0 for moderate activity, and 3.3 for walking) by the minutes of activity per day and the number of days per week. The total MET score is obtained by summing the MET-minutes per week from all activity levels [14]. The Korean version of the IPAQ-SF has also been cross-culturally adapted and validated to ensure its reliability and applicability in Korean clinical settings [15].

### Statistical analyses

Descriptive statistics were performed to evaluate mean, standard deviation (SD) and proportion. Sample size adequacy was determined based on the guidelines by Bujang (2024), which recommend a minimum of 250 participants to detect moderate correlations ($r \approx 0.3$) with sufficient statistical power [16,17]. Kolmogorov-Smirnov test was conducted to ensure the normality of the dataset. Multivariable partial-correlation analyses were performed to assess relationships between MET-minutes and clinical parameters after controlling for key covariates including age, sex, BMI, pain severity (VAS). Multivariable linear regression models were fitted for each FAOS subscale (Symptoms, Pain, Activities of Daily Living, Sports/Recreation, and Quality of Life) including age, sex, BMI, and pain severity (VAS) as covariates. Multicollinearity among covariates was assessed using variance inflation factors (VIF), with VIF < 5 considered acceptable, and no concerning collinearity was observed (all VIF ≤ 1.017). Pain severity (VAS) was included a priori as a covariate because pain is a key clinical determinant of functional limitation and may confound the association between osteoarthritis severity and physical activity. To adjust for multiple comparisons, the Holm–Bonferroni correction was applied to control the false discovery rate (FDR). Adjusted p-values are reported, and statistical significance was set at p < 0.05 after correction. All statistical analyses were performed using SPSS software (version 29.0; IBM, Armonk, New York, United States).

### Results

A total of 262 patients with ankle osteoarthritis were included in the final analysis (Fig 1). Baseline characteristics are shown in Table 1, and sex- and Takakura stage–stratified MET-min/week summary statistics (mean ± SD and median [IQR]) are provided in Supporting information table 1 (S1 Table). There were 99 males and 163 females, and mean age of the patients was 66.8 years (SD 8.7 years). Mean BMI was 26.3 kg/m$^2$ (SD 3.4 kg/m$^2$) and Takakura stage was assigned as follows: 54 patients in stage 2, 93 in stage 3a, 52 in stage 3b, and 63 in stage 4. Mean total MET-minutes was 2021.6 (SD 3476.6) (Table 1).

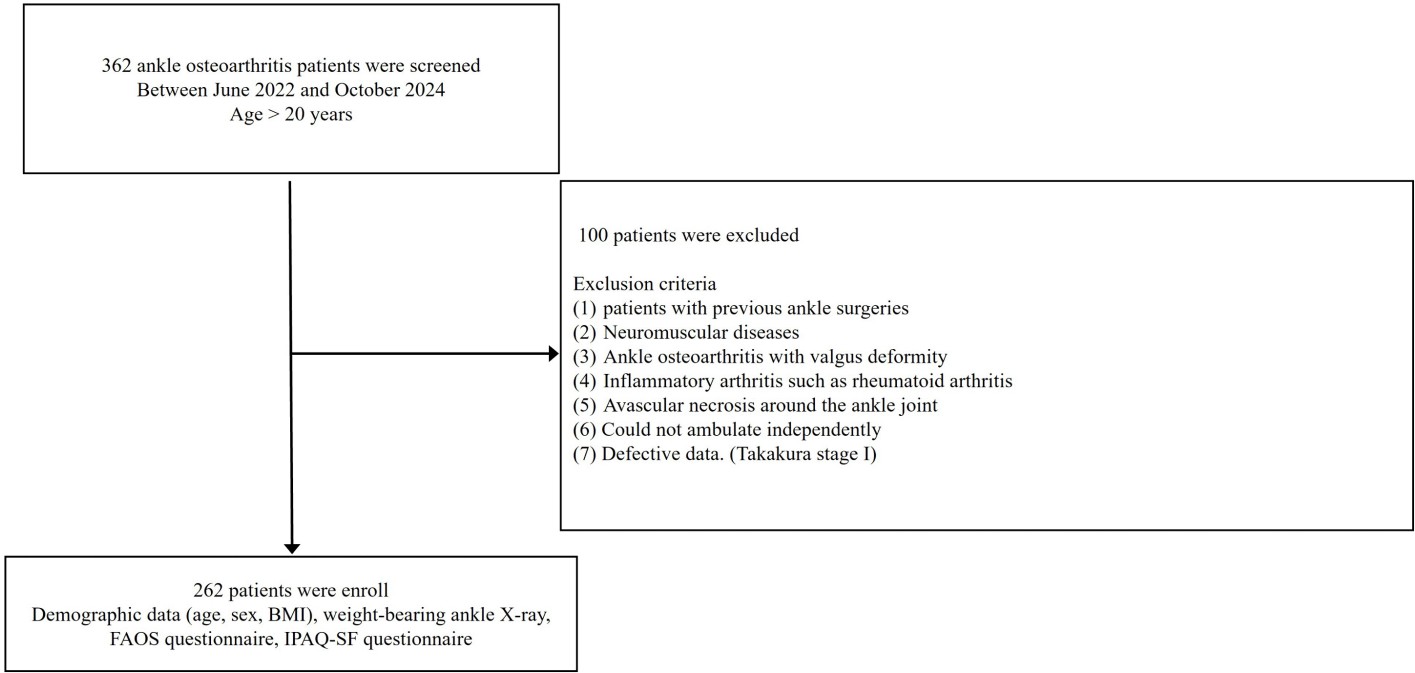

**Fig 1. Flowchart of the inclusion and exclusion criteria.**

**Table 1. Patients demographics.**

| Parameter | Value | Median [IQR] |
|---|---|---|
| Age (years ± SD) | 66.8 (SD 8.7) | |
| Sex (M/F) | 99/ 163 | |
| BMI (kg/m²) | 26.3 (SD 3.4) | |
| Takakura stage (2/3a/3b/4) | 54/ 93/ 52/ 63 | |
| FAOS | | |
| Symptom | 66.8 (SD 19.6) | |
| Pain | 61.8 (SD 19.9) | |
| ADL | 68.7 (SD 19.8) | |
| Sports | 38.5 (SD 25.7) | |
| QoL | 35.6 (SD 20.9) | |
| Pain VAS | 5.7 (SD 2.4) | |
| IPAQ (I/ II/ III) | 206/ 37/ 19 | |
| Total MET-minutes | 2021.6 (SD 3476.6) | 924 [297–2438] |
| Vigorous activity MET-minutes | 404.4 (SD 2422.8) | 0 [0–0] |
| Moderate activity MET-minutes | 539.9 (SD 1280.3) | 0 [0–640] |
| Walking MET-minutes | 1077.3 (SD 1587.4) | 594 [198–1287] |

SD = standard deviation; M = male; F = female; FAOS = Foot and Ankle Outcome Score; IPAQ = International Physical Activity Questionnaire; MET = Metabolic Equivalent Task minutes.

Comparisons between males and females were performed using the independent t-test for continuous variables, and the chi-square test for categorical variables. Moderate MET-minutes (p=0.033) and walking MET-minutes (p=0.035) were significantly different between male and female patients. Age was negatively correlated with vigorous activity MET-minutes (r=−0.155, p=0.010) and total MET-minutes (r=−0.182, p=0.003). BMI was significantly correlated with vigorous activity MET-minutes (r=−0.129, p=0.038), walking MET-minutes (r=−0.141, p=0.024) and total MET-minutes (r=−0.177, p=0.005).

To improve clarity, we present overall adjusted associations in Table 2 and sex-stratified adjusted associations in Tables 3 and 4. In a whole patient group, age was significantly correlated with vigorous MET-minutes (r=−0.195, p=0.002) and total MET-minutes (r=−0.182, p=0.003). BMI was significantly correlated with vigorous MET-minutes (r=−0.129, p=0.038), walking MET-minutes (r=−0.141, p=0.024) and total MET-minutes (r=−0.177, p=0.005). Pain VAS was not significantly correlated with MET-minutes. Takakura stage was significantly correlated with vigorous MET-minutes (r=−0.152, p=0.016) and total MET-minutes (r=−0.127, p=0.044) controlled for age, gender, BMI and pain VAS (Table 2).

In male patient group, MET-minutes was not correlated with age, BMI or pain VAS. Takakura stage was significantly correlated with vigorous MET-minutes (r=−0.226, p=0.027) controlled for age, BMI, and pain VAS (Table 3). In female patient group, age was significantly correlated with vigorous MET-minutes (r=−0.228, p=0.004), walking MET-minutes (r=−0.197, p=0.013) and total MET-minutes (r=−0.246, p=0.002). BMI was significantly correlated with vigorous MET-minutes (r=−0.167, p=0.036), walking MET-minutes (r=−0.188, p=0.018) and total MET-minutes (r=−0.198,

**Table 2. Partial correlations controlled for age, gender, BMI and pain VAS in a whole patient group.**

|  | Takakura | Vigorous_MET | Moderate_MET | Walking_MET |
|---|---|---|---|---|
| Vigorous_MET | −0.152*<br>(p=0.016) |  |  |  |
| Moderate_MET | 0.093<br>(p=0.141) | −0.008<br>(p=0.896) |  |  |
| Walking MET | −0.118<br>(p=0.060) | 0.203*<br>(p=0.001) | 0.044<br>(p=0.486) |  |
| Total_MET | −0.127*<br>(p=0.044) | 0.797*<br>(p<0.001) | 0.394*<br>(p<0.001) | 0.621*<br>(p<0.001) |

MET=Metabolic Equivalent Task minutes.

* p<0.05; adjusted for multiple comparisons using Holm-Bonferroni method.

**Table 3. Partial correlations controlled for age, BMI and pain VAS in male patient group.**

|  | Takakura | Vigorous_MET | Moderate_MET | Walking_MET |
|---|---|---|---|---|
| Vigorous_MET | −0.226*<br>(p=0.027) |  |  |  |
| Moderate_MET | 0.174<br>(p=0.091) | −0.023<br>(p=0.825) |  |  |
| Walking_MET | −0.099<br>(p=0.339) | 0.264*<br>(p=0.009) | 0.037<br>(p=0.718) |  |
| Total_MET | −0.050<br>(p=0.627) | 0.523*<br>(p<0.001) | 0.568*<br>(p<0.001) | 0.765*<br>(p<0.001) |

MET=Metabolic Equivalent Task minutes.

* p<0.05; adjusted for multiple comparisons using Holm-Bonferroni method.

**Table 4. Partial correlations controlled for age, BMI and pain VAS in female patient group.**

| | Takakura | Vigorous_MET | Moderate_MET | Walking_MET |
|---|---|---|---|---|
| Vigorous_MET | −0.144 (p=0.074) | | | |
| Moderate_MET | −0.003 (p=0.973) | 0.015 (p=0.854) | | |
| Walking_MET | −0.143 (p=0.076) | 0.235* (p=0.003) | 0.051 (p=0.532) | |
| Total_MET | −0.167* (p=0.037) | 0.897* (p<0.001) | 0.296* (p<0.001) | 0.553* (p<0.001) |

MET=Metabolic Equivalent Task minutes.

* p<0.05; adjusted for multiple comparisons using Holm-Bonferroni method.

p=0.013). Pain VAS was not significantly correlated with MET-minutes. After controlling for age, BMI and pain VAS, Takakura stage was significantly correlated with total MET-minutes (r=−0.167, p=0.037) (Table 4).

In multivariable models adjusting for age, sex, and BMI, higher pain severity (VAS) was consistently associated with worse scores across all FAOS subscales (all p<0.001). Subsequently, stage-stratified analyses showed the following associations between FAOS subscales and MET-min/week (S2–S5 Tables).

Detailed stage-stratified correlation results are provided in S2–S5 Tables (Supporting Information). In patients with Takakura stage 2, BMI showed significant negative correlation with vigorous MET-minutes (r=−0.282, p=0.039) and total MET-minutes (r=−0.305, p=0.025). The symptom subscale of FAOS was negatively correlated with vigorous MET-minutes, walking MET-minutes and total MET-minutes. The pain subscale of FAOS was negatively correlated with total MET-minutes (S2 Table).

In patients with Takakura stage 3a, the symptom subscale of FAOS was positively correlated with vigorous MET-minutes (r=0.242, p=0.020) (S3 Table). In Takakura stage 3b, vigorous MET-minutes was negatively correlated with age (r=−0.325, p=0.019) and pain VAS (r=−0.283, p=0.042). The ADL subscale and pain subscale of FAOS were correlated with vigorous MET-minutes and total MET-minutes (S4 Table).

In patients with Takakura stage 4, there was no patients doing vigorous activities. The sports subscale and QoL subscale of FAOS was significantly correlated with Walking MET-minutes (S5 Table).

## Discussion

The study findings highlighted that physical activity levels, as measured by MET-minutes, were associated with radiographic stage, demographic characteristics, and various aspects of subjective symptoms in patients with ankle osteoarthritis. These associations were of limited strength and should be interpreted as observational rather than causal. These results are broadly consistent with prior studies in knee and hip osteoarthritis reporting lower physical activity with greater disease burden, although patterns may differ by joint and patient characteristics. The hypothesized pathway linking ankle osteoarthritis severity, reduced physical activity, and potential cardiometabolic implications (Fig 2). Although reduced physical activity is plausibly linked to adverse cardiometabolic health, metabolic outcomes were not directly measured in this study. Therefore, our findings should be interpreted as showing an association between radiographic ankle osteoarthritis severity and lower self-reported physical activity, and the potential metabolic implications remain hypothesis-generating and require confirmation in future longitudinal studies incorporating cardiometabolic endpoints.

In the female cohort, age and BMI were found to be negatively correlated with physical activity levels in ankle OA. Both increased age and higher BMI are well-established risk factors for metabolic diseases [18]. Of the two, a higher BMI is considered to be an important modifiable risk factor in the development of metabolic diseases as well as for reduced

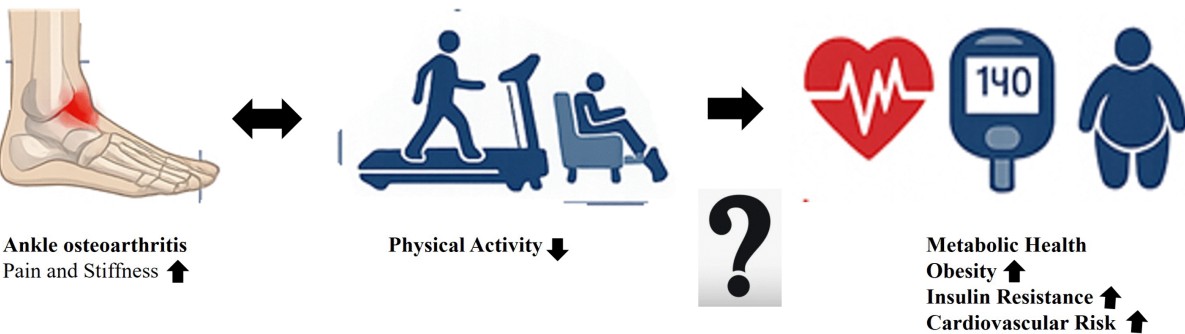

**Fig 2. Conceptual diagram illustrating the hypothesized pathway linking radiographic ankle osteoarthritis severity to reduced physical activity and potential cardiometabolic implications.** Greater ankle osteoarthritis severity may be associated with increased pain and stiffness and reduced mobility, which may contribute to lower habitual physical activity (MET-min/week). Reduced physical activity may in turn be related to decreased energy expenditure and fitness, with potential downstream associations with adverse cardiometabolic health outcomes (e.g., obesity, insulin resistance, and cardiovascular risk).

physical activity in patients with ankle osteoarthritis [19]. Obesity not only increases mechanical stress on joints but also triggers metabolic changes that exacerbate inflammation, making it a significant risk factor for both the development and progression of osteoarthritis and metabolic diseases such as type 2 diabetes and cardiovascular disease [20,21]. Maintaining physical activity may help mitigate the effect of chronic low grade inflammation, which could be beneficial in preventing the progression of both osteoarthritis and metabolic diseases [22–25]. However, the detailed optimization of the intensity and amount of physical activity for each patient should be further researched. Although we present sex-stratified findings for both males and females, associations were more consistent in females in our dataset; therefore, interpretations remain cautious and subgroup findings should be considered exploratory given potential differences in sample size and statistical power.

The relationship between subjective qualitative assessments of activity (such as FAOS ADL and sports subscales) and semiquantitative activity levels (IPAQ and MET) did not necessarily correlate with each other. Subjective assessments, such as those captured by the FAOS, may be influenced by individual perception, recall bias, and personal interpretation of pain and limitations. Tzetzis et al. indicated that subjective evaluations often overestimate activity duration and underestimate activity intensity, highlighting discrepancies between perceived and actual activity levels [26]. Wainwright and Kehlet noted that post-rehabilitation outcomes in hip or knee arthroplasty often show improvement in patient-reported outcomes, even when objective measures do not reflect such gains [27]. This underscores the importance of integrating both subjective and objective assessments to obtain a comprehensive understanding of patient health, ultimately guiding personalized treatment strategies more effectively [28].

In patients with Takakura stage 2, increased physical activity levels were associated with worsening symptoms (symptom subscale) and increased pain (pain subscale), whereas lower pain levels and better quality of life were correlated with increased physical activity in those with Takakura stage 3b. This inconsistent relationship between patients' subjective symptoms and physical activity across different stages of ankle OA may be due to a stage-specific anti-inflammatory effect of physical activity or confounding factors, such as differences in medical treatment distributions across patient stages. More detailed biochemical research focusing on the impact of physical activity levels is required to clarify this issue.

For patients in Takakura stage 4 (end-stage ankle arthritis), none engaged in vigorous activity. This suggests that end-stage ankle osteoarthritis severely restricts patient activity levels, leading to reduced MET-minutes, which may, in turn, contribute to additional systemic health complications. High-intensity exercise has been reported to be more effective in preventing metabolic diseases compared to low-intensity exercise [29]. Therefore, this patients group could be at a

specific risk for developing metabolic diseases. Currently, the goal of treatment for ankle OA is focused on reducing pain, but a different treatment strategy might be required considering the activity limitation and the consequent medical problems for this advanced group of patients [30]. The impact of ankle OA treatment on physical activity levels and the associated internal medical effects warrants further prospective longitudinal research.

This study has several limitations. First, the retrospective cross-sectional design does not allow inference of causality or directionality between ankle osteoarthritis severity and physical activity. Second, the tertiary referral setting and the absence of a healthy control or other-arthritis comparator may introduce selection bias and limit generalizability and contextual interpretation. In addition, cardiometabolic comorbidities and metabolic outcomes were not systematically assessed; thus, any cardiometabolic implications should be considered hypothesis-generating. Third, physical activity was assessed using the self-reported IPAQ-SF and may be affected by recall and social desirability bias; moreover, MET-min/week variables were right-skewed with many zero values (especially vigorous activity). In Takakura stage 4, vigorous activity was uniformly absent, creating a floor effect; therefore, total MET-min/week (walking + moderate activity) was used to represent overall activity volume in this subgroup, although intensity-specific associations could not be evaluated. Accordingly, correlation-based estimates may have been influenced by non-normality, zero-inflation, and outliers. Fourth, while Holm–Bonferroni adjustment was applied within each correlation table, multiple related subgroup and exploratory analyses were conducted across several tables; thus, stage-stratified findings should be interpreted as exploratory given the possibility of residual multiplicity. Finally, the effect of medical treatment was not considered and may have influenced FAOS scores. Because pain, treatment exposure, and physical activity may interact in a bidirectional and time-dependent manner, the retrospective cross-sectional design cannot determine temporal sequence and residual confounding may remain. Future prospective or case-control studies using objective activity measures (e.g., accelerometers) and cardiometabolic endpoints are warranted to reduce self-report bias and better evaluate the directionality of observed associations.

## Conclusion

In this single-center, retrospective cross-sectional study, self-reported physical activity (MET-min/week) was associated with radiographic ankle osteoarthritis severity and demographic factors such as age and BMI, with greater radiographic severity corresponding to lower physical activity. These findings represent correlations and do not establish cause-and-effect relationships or directionality. Because pain, treatment exposure, and activity may interact in a time-dependent and bidirectional manner, future prospective longitudinal studies with repeated assessments and objective activity measures are needed to clarify temporal relationships and clinical implications.

## Supporting information

**S1 Table. Table 1–1 (male), Table 1–2 (female) MET-min/week summary statistics (mean ± SD and median [IQR]) stratified by sex and Takakura stage.**
(DOCX)

**S2 Table. Correlation table between factors in Takakura stage 2.**
(DOCX)

**S3 Table. Correlation table between factors in Takakura stage 3a.**
(DOCX)

**S4 Table. Correlation table between factors in Takakura stage 3b.**
(DOCX)

**S5 Table. Correlation table between factors in Takakura stage 4.**
(DOCX)

## Acknowledgments

The authors would like to express their gratitude to all participants and collaborators who contributed to this study. We also appreciate the support from our colleagues at Seoul National University College of Medicine for their valuable insights and technical assistance.

## Author contributions

**Conceptualization:** Dong Yeon Lee, Kyoung Min Lee.

**Data curation:** Hee-jin Yang, Hee Soo Han, Dong Yeon Lee, Kyoung Min Lee.

**Formal analysis:** Woo Sub KIM.

**Investigation:** Ji Hye Choi.

**Methodology:** Ji Hye Choi, Hee Soo Han, Kyoung Min Lee.

**Resources:** Hee-jin Yang.

**Supervision:** Kyoung Min Lee.

**Writing – original draft:** Woo Sub KIM.

**Writing – review & editing:** Woo Sub KIM.

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
