## [Decision Letter · Decision Letter 0]

16 Jan 2026

PONE-D-25-50381Relationship Between Physical Activity and Ankle Osteoarthritis: Implication as a potential candidate risk factor for metabolic diseasesPLOS One

Dear Dr. Lee,

Thank you for submitting your manuscript to PLOS ONE. After careful consideration, we feel that it has merit but does not fully meet PLOS ONE’s publication criteria as it currently stands. Therefore, we invite you to submit a revised version of the manuscript that addresses the points raised during the review process.

**ACADEMIC EDITOR: Please make necessary corrections based on the comments provided by the reviewers.**

We look forward to receiving your revised manuscript.

Kind regards,

Zulkarnain Jaafar

Academic Editor

PLOS One

Journal Requirements:

3. We note that your Data Availability Statement is currently as follows: If the data are all contained within the manuscript and/or Supporting Information files, enter the following: All relevant data are within the manuscript and its Supporting Information files.

Reviewers' comments:

Reviewer's Responses to Questions

**Comments to the Author**

1. Is the manuscript technically sound, and do the data support the conclusions?

Reviewer #1: Yes

Reviewer #2: Partly

2. Has the statistical analysis been performed appropriately and rigorously? 

Reviewer #1: Yes

Reviewer #2: No

3. Have the authors made all data underlying the findings in their manuscript fully available?

Reviewer #1: Yes

Reviewer #2: No

4. Is the manuscript presented in an intelligible fashion and written in standard English?

Reviewer #1: Yes

Reviewer #2: Yes

5. Review Comments to the Author

Reviewer #1: The manuscript investigates relationship between physical activity (measured via MET-minutes) and ankle osteoarthritis (OA) severity, with implications for metabolic disease risk. It is a retrospective cross-sectional study that informs how OA-related activity limitations may metabolic systemic health.

Concerns:

• Limitations to be acknowledged in the discussion or remedied: As a cross section study the observed associations do not establish directionality. There is a risk of selection bias as the study was conducted at a tertiary referral center, the cohort may represent more severe cases, limiting generalizability. The self-reported physical activity is subject to recall and social desirability bias. There is no comparison with healthy controls or patients with other types of arthritis limits context for the findings.

• Some of the terminologies are inconsistent: some tables use “light activity” while IPAQ-SF typically uses “walking”; clarity is needed.

• Was the MET data normally distributed? The MET-minute data (especially vigorous activity) mean total MET = 2021.6, SD = 3476.6 suggests right skew. Were transformations or robust methods considered?

• Please clarify how multiple testing was handled across all subgroup analyses (Tables 2–8).

• Provide MET summary statistics by Takakura stage and gender in supplementary materials.

• Missing Key Descriptive Stats for METs by Subgroup: Mean ± SD or median (IQR) for MET-minutes by gender and Takakura stage are not shown in Table 1, only overall. A supplementary table with METs stratified by stage and sex can resolve this issue.

• In Takakura stage 4, no patients reported vigorous activity, making correlations impossible. This is noted, but the analysis could be refined—e.g., combining moderate + walking METs for that group.

• No Adjustment for Multiple Testing Across All Tables. While Holm–Bonferroni was applied within tables, there are 8 tables total, increasing the family-wise error rate across the study. Suggestion: Acknowledge this in limitations or apply a stricter correction across all primary outcomes.

• Pain VAS was not correlated with MET-minutes in many groups, yet it was included in all models. The rationale should be stated.

• No mention of checking for multicollinearity (e.g., VIF) or linearity assumptions in partial correlations.

Suggestions:

• Introduction & Discussion: State explicitly the hypothesized mechanistic pathway between ankle OA → reduced PA → metabolic disease risk. Discuss how findings align or contrast with existing literature on knee/hip OA and metabolic health.

• Improve Data Presentation: Correct table typos (“Thakstan” vs. “Takakura”) and ensure consistent terminology. Consider consolidating some correlation tables or moving them to supplementary materials to improve readability. For example: Tables 2–4 (Partial correlations by whole group, male, female) could be consolidated into one table with clear column labeling for subgroups. Tables 5–8 (Correlations by Takakura stage) could either be combined into one summary table highlighting only significant correlations, or moved to Supplementary Materials.

• Expand in discussion of limitation on how future studies could overcome the limitations of self-reporting (e.g., using accelerometers). A prospective or case-control design could better establish causality.

• Refine the Conclusion: Make the “take-home” message clearer regarding clinical implications. Should ankle OA management include PA promotion to mitigate metabolic risk?

• Consider a Visual Abstract or Summary Figure: A conceptual diagram linking ankle OA, PA, and metabolic disease could enhance reader engagement and clarity.

• Some references (e.g., #6 on cancer) seem misplaced; ensure all citations directly support the manuscript’s claims.

Reviewer #2: Introduction

Line 60: Expand the statements on ankle osteoarthritis and what is the cause of this pathology.

Line 77: Expand the study objective. The pathology and physical activity can be more detailed. As written it appears yes/no for pathology.

Methods

Line 92: The authors discuss the disease state that can result from lack of activity brought on by ankle OA. Other medical history data would support the objective of the study to determine the systemic level of disease.

Line 96: While trying to maintain a homogenous sample, the exclusion criteria appear to have potential impact on the results. Why eliminate variables that would impact physical activity and other severities? there are also subsets of patients that could be analyzed and provide a larger impact on the results.

Line 151: The statistical analysis is very basic. There are other questions that can asked of the data to beyond correlations. Consider stratification based on the clinical measures.

Results

Line 164: Provide data mean (sd) in table to support the correlations. Relevant data are not provided and would allow the reader to better interpret the findings.

Line 166: Data are provided for total enrollment of males v females but no other stratification of the other variables. If sex is a consideration the results should be stated for overall, males, and females. Also if this relevant, comparisons data should be available.

Line 170: What analyses were used for comparisons reference between males and females for Met-minutes to determine differences existed? Why not include group comparisons for sex. Sex is not identified in the introduction as a key factor for the proposed relationship.

Line 175: How are these findings different from those starting at line 170? The previous paragraph is not specific to sex but appears to be total. Starting at 175 the whole group is also referenced so not clear.

Discussion

Line 206: This findings are overgeneralized. While many of the correlations were significant, they are weak.

Line 211: Why only discuss the female cohort.

Line 218: With reference to system diseases, this information is not presented. Recommend extracting from the medical record and including in the analyses.

Conclusion

Line 267: The conclusion is overstated and only focused on correlations. While significant relationships exist, they are weak. Other data should be included in the analyses to better support the conclusion.

6. PLOS authors have the option to publish the peer review history of their article (what does this mean?). If published, this will include your full peer review and any attached files.

Reviewer #1: No

Reviewer #2: No

---

## [Author Response · Author response to Decision Letter 1]

28 Feb 2026

We sincerely thank the Academic Editor and reviewers for their thoughtful and constructive comments. We have carefully revised the manuscript in response to all points raised. A detailed, point-by-point response outlining the changes made is provided in the accompanying response document.

---

## [Decision Letter · Decision Letter 1]

22 Apr 2026

Relationship Between Physical Activity and Ankle Osteoarthritis: Implications for metabolic diseases

PONE-D-25-50381R1

Dear Dr. Lee,

We’re pleased to inform you that your manuscript has been judged scientifically suitable for publication and will be formally accepted for publication once it meets all outstanding technical requirements.

Kind regards,

Zulkarnain Jaafar

Academic Editor

PLOS One

Additional Editor Comments (optional):

Reviewers' comments:

Reviewer's Responses to Questions

**Comments to the Author**

1. If the authors have adequately addressed your comments raised in a previous round of review and you feel that this manuscript is now acceptable for publication, you may indicate that here to bypass the “Comments to the Author” section, enter your conflict of interest statement in the “Confidential to Editor” section, and submit your "Accept" recommendation.

Reviewer #1: All comments have been addressed

Reviewer #2: All comments have been addressed

2. Is the manuscript technically sound, and do the data support the conclusions?

Reviewer #1: Yes

Reviewer #2: Yes

3. Has the statistical analysis been performed appropriately and rigorously? 

Reviewer #1: Yes

Reviewer #2: Yes

4. Have the authors made all data underlying the findings in their manuscript fully available?

Reviewer #1: Yes

Reviewer #2: Yes

5. Is the manuscript presented in an intelligible fashion and written in standard English?

Reviewer #1: Yes

Reviewer #2: Yes

6. Review Comments to the Author

Reviewer #1: (No Response)

Reviewer #2: All comments have been addressed by the authors and the manuscript now reads clearer with the additional information. Thank you for your attention.

7. PLOS authors have the option to publish the peer review history of their article (what does this mean?). If published, this will include your full peer review and any attached files.

Reviewer #1: No

Reviewer #2: No

---

## [Editor Report · Acceptance letter]

PONE-D-25-50381R1

PLOS One

Dear Dr. Lee,

I'm pleased to inform you that your manuscript has been deemed suitable for publication in PLOS One. Congratulations! Your manuscript is now being handed over to our production team.

Kind regards,

on behalf of

Dr. Zulkarnain Jaafar

Academic Editor

PLOS One